# BEYOND SIGHT: PROBING ALIGNMENT BETWEEN IMAGE MODELS AND BLIND V1

**Galen Pogoncheff**[1†*], **Jacob Granley**[1†*], **Alfonso Rodil**[2], **Leili Soo**[2], **Lily Marie Turkstra**[3],
**Lucas Gil Nadolskis**[4], **Arantxa Alfaro Saez**[2], **Cristina Soto Sanchez**[2],
**Eduardo Fernandez Jover**[2], **& Michael Beyeler**[1, 3]

## ABSTRACT

Neural activity in the visual cortex of blind humans persists in the absence of visual stimuli. However, little is known about the preservation of visual representation capacity in these cortical regions, which could have significant implications for neural interfaces such as visual prostheses. In this work, we present a series of analyses on the shared representations between evoked neural activity in the primary visual cortex (V1) of a blind human with an intracortical visual prosthesis, and latent visual representations computed in deep neural networks (DNNs). In the absence of natural visual input, we examine two alternative forms of inducing neural activity: electrical stimulation and mental imagery. We first quantitatively demonstrate that latent DNN activations are aligned with neural activity measured in blind V1. On average, DNNs with higher ImageNet accuracy or higher sighted primate neural predictivity are more predictive of blind V1 activity. We further probe blind V1 alignment in ResNet-50 and propose a proof-of-concept approach towards interpretability of blind V1 neurons. The results of these studies suggest the presence of some form of natural visual processing in blind V1 during electrically evoked visual perception and present unique directions in mechanistically understanding and interfacing with blind V1.

## 1 INTRODUCTION

Despite lacking visual input, neural activity in the visual cortex of blind humans persists after vision loss (Burton, 2003; Fine et al., 2003; Huber et al., 2015; Striem-Amit et al., 2015; Fine & Park, 2018). Research suggests that cortical plasticity enables the cortex to undertake non-visual tasks (Burton et al., 2002; Sadato et al., 2002), though the extent to which visual processing capabilities are preserved remains under explored (Lambert et al., 2004; Striem-Amit et al., 2012; Hahamy et al., 2021). Unraveling how the blind visual cortex represents visual information could unlock profound insights into brain function, cortical plasticity, and sensory compensation (Beyeler et al., 2017). This knowledge is crucial for advancing rehabilitative technologies, like visual neuroprostheses, which evoke visual sensations (*phosphenes*) through electrical stimulation (Fernandez, 2018).

Recent studies have shown that the latent representations in DNNs trained for vision tasks align with neural activity in the visual cortex of sighted primates (Yamins et al., 2013; 2014; Cadena et al., 2019; Schrimpf et al., 2020; Marques et al., 2021; Pogoncheff et al., 2023). This synergy has fueled advancements in visual neuroscience, such as the discovery of maximally exciting images for individual neurons in primate cortex (Bashivan et al., 2019). Yet, the critical question remains: does this alignment hold for spiking activity in visual cortex of humans or, specifically, the blind? Answering this could pave the way for accurate modeling of visual processing in the blind human cortex, enabling the design of prostheses which deliver more perceptually intelligible visual experiences (de Ruyter van Steveninck et al., 2022; Granley et al., 2022a;b; 2023; Castro et al., 2023).

[1]Department of Computer Science, University of California, Santa Barbara, CA, USA
[2]Instituto Bioingeniería, Universidad Miguel Hernández, Elche, Spain
[3] Department of Psychological and Brain Sciences, University of California, Santa Barbara, CA, USA
[4] Department of Dynamical Neuroscience, University of California, Santa Barbara, CA, USA
*These authors contributed equally
†Corresponding author: {galenpogoncheff,jgranley}@ucsb.edu

In this study, we analyze the alignment of DNN representations with neural activity from an awake blind human participant implanted with a 96 channel intracortical prosthesis (Fernández & Normann, 2017) measured during a delayed-response working memory (WM) task (Figure 1). Neural activity related to visual WM has been found in primates and humans (Albers et al., 2013), but the role of spiking activity in human V1 during WM to support visual processing is not fully understood. This setup therefore provides a unique opportunity to probe the similarities between DNN-computed visual representations and human perceptual representations. Novel contributions include:

- We show a significant representational alignment between DNN activations and V1 in a blind human for both spiking activity and local field potentials. Across 69 DNNs, blind V1 alignment was positively correlated with 1) DNN alignment with neural activity in the visual cortex of sighted primates as well as 2) DNN ImageNet object recognition accuracy.
- We demonstrate that DNN activity not only aligns with neural activity during stimulation, but also with local field potentials measured after stimulation or during mental imagery.
- We propose a proof-of-concept approach towards enhancing the interpretability of neurons recorded in blind V1, a neural system which cannot be probed with traditional psychophysical experiments that rely on the ability to present visual stimuli (e.g., image or video).

These preliminary results mark a crucial step towards understanding visual processing in the blind visual cortex—a key to advancements in both neuroscience and rehabilitative neuroengineering.

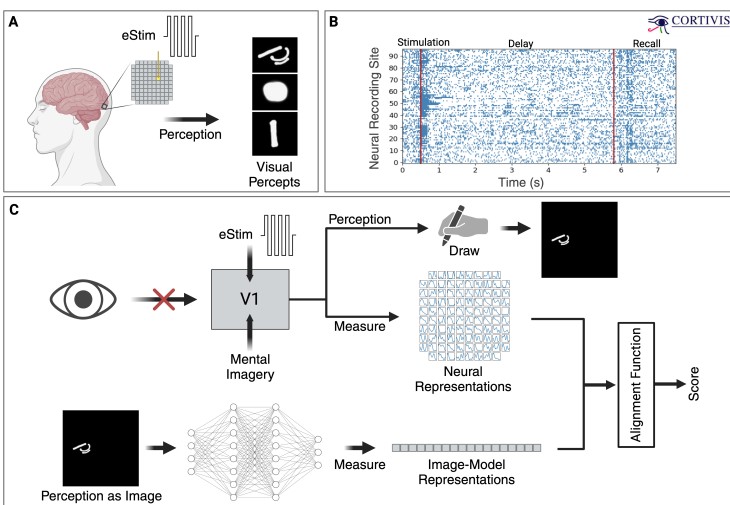

Figure 1: Schematic of neural recording and alignment evaluation. **A**. Electrical stimulation delivered via an intracortical prosthesis (Cortivis) elicits visual perception. **B**. Intracortical neural recordings are simultaneously acquired during stimulation, delay, and recall periods of a working memory task. **C**. Representational alignment between DNN and blind V1 are evaluated by comparing representations measured from both systems when presented with the same perceptual input.

## 2 METHODS

### 2.1 EVOKING VISUAL PERCEPTION VIA INTRACORTICAL ELECTRICAL STIMULATION

The data analyzed in this work was acquired using recordings and behavioral responses from an awake blind human with a 96-channel intracortical visual prosthesis (Fernandez & Normann, 2017), implanted in right parafoveal V1, during a delayed-response WM task. The participant (61, Male) had no light perception for 5 years due to optic neuropathy following pituitary adenoma surgery. Over 90 trials, one of three chosen electrodes was stimulated with a $167\,\text{ms}$ cathodic-first biphasic pulse train, reliably evoking a spatially distinct phosphene ("stimulation period", Figure 1A). The participant would then remember the evoked visual perception during a $3-5\,\text{s}$ "delay" period, after which an audio cue triggered the participant to visually recollect the phosphene in vivid detail (accounting for shape, brightness, and visual field location; "recall period"). Following ten consecutive

trials for a given electrode, the participant was asked to verbally describe and draw the perceived phosphene. These drawings were then refined by a sighted researcher during an extensive discussion with the participant, given the difficulty of replicating the perceived phosphene without vision.

## 2.2 Data Processing

Each phosphene drawing was digitized and placed within a $244 \times 244$ pixel image with black background, assumed to span 8 degrees visual angle (DVA). Based on verbal feedback from the participant, the phosphene drawings were scaled to span 2 DVA and placed within the left visual field of the image (corresponding to the implant's placement in the right occipital lobe). Noise in raw neural activity resulting from stimulation artifacts was suppressed via blanking during the stimulation window and interpolation with a 3rd order polynomial. Multi-unit activity (MUA), entire/envelope spiking activity (ESA), as well as $\theta$, $\alpha$, $\beta$, and $\gamma$ bands of the local field potential (LFP) were extracted from the filtered neural activity (Appendix A.3).

## 2.3 Evaluating Representational Alignment

Traditionally, the representational alignment between two systems is measured by inputting the same stimulus to each system, measuring an internal state for each system (e.g., DNN activations, neural activity), optionally projecting the latent activity to a shared space, and quantifying the similarity between embeddings with an alignment score (Sucholutsky et al., 2023). Common techniques include representational similarity analysis (RSA) (Kriegeskorte et al., 2008) and linear regression followed by a correlational score (Schrimpf et al., 2020; Cadena et al., 2019; Conwell et al., 2022).

For blind subjects, however, there is no natural visual input. Therefore, we investigated neural activity triggered by electrical stimulation or mental imagery. To obtain DNN activations, we used the corresponding phosphene drawings as input, assessing the network's capability to mirror the neural activity associated with the participant perceiving said phosphene (Fig. 1C). Importantly, these networks were not designed to predict neural activity, but to perform ImageNet classification.

To quantify representational alignment between DNN and blind V1 activity, we utilized RSA and correlations of learned linear encodings. For RSA, correlations between trials of neural activity were used to compute dissimilarity matrices, which were compared using correlation score. For linear regression, cross-validated ridge regression from PCA-projected model activity was used to predict biological neural activity measured on each of 96 channels. The alignment between neural representations and encoded DNN representations was then scored according to Pearson's correlation coefficient (Schrimpf et al., 2020).

## 3 Results

### 3.1 Sustained stimulus-selective multi-unit activity in human V1

We first examined how intracortical recordings differed across the three phases of the WM task. We found significant increases in neural activity at the onset of stimulation and the audio cue for recall initiation (t-test, $p < 0.001$). During stimulation, delay, and recall periods, neural activity was significantly different than spontaneous activity ($p < 0.001$). Analysis revealed shared neural activity patterns per electrode across stimulation, delay, and recall periods, allowing for the successful classification of stimulating electrodes based on recall-period activity alone, achieving 97% accuracy for the delay and 88% for the stimulation periods using a random forest classifier. These shared signatures could be learned from the delay or recall period activity, but not from the electrically evoked activity, suggesting that WM elicits a subset of the full activity present during electrical stimulation.

### 3.2 DNN-Sighted V1 Alignment Correlates with DNN-Blind V1 Alignment

We then examined how well different brain-like DNNs aligned with intracortical recordings in blind V1. We selected 69 models evaluated on the Brain-Score benchmark suite (Schrimpf et al., 2020) as candidates for measuring DNN-Blind V1 alignment. DNN-blind V1 alignment was measured for all six feature sets (MUA, ESA, and LFP measures) during the stimulation (Figure 2, *top*) and recall

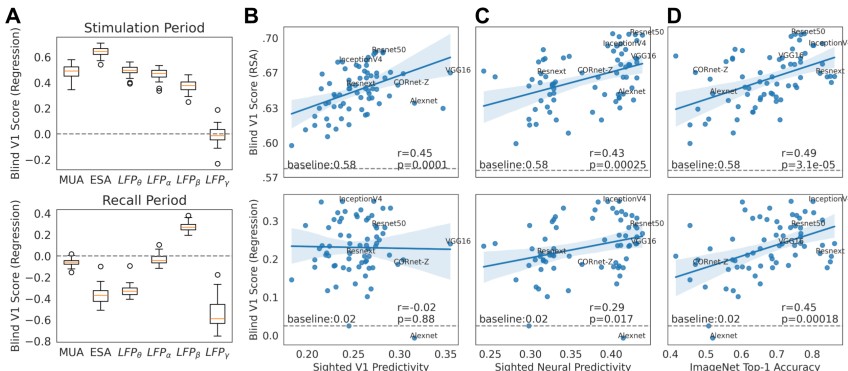

Figure 2: **A.** Distribution of blind V1 alignment scores (regression method) across DNNs of different features for stimulation and recall activity. **B-D.** Scatterplots showing correlation between sighted V1 predictivity (B), overall sighted neural predictivity (C) or ImageNet top 1 accuracy (D) with blind MUA representational similarity score (top: RSA, bottom: regression) across DNNs for stimulation-evoked activity. Dashed lines: baseline alignment measured without DNN features.

period (*bottom*). A baseline similarity score (*dashed lines*) measures the representational alignment (using regression or RSA) calculated directly from the drawing, without using DNN activations.

For electrically evoked activity, blind V1 alignment scores were significantly higher than baselines (t-test, $p < 0.001$) for all features except LFP$_\gamma$. For recall activity, only LFP$_\beta$ activity predictivity was significantly higher than the baseline. Across DNNs, we found a significant, moderate positive correlation between blind V1 scores and sighted V1 predictivity scores (Figure 2B; Freeman et al., 2013), sighted neural predictivity across V1, V2, V4, and IT (Figure 2C; Schrimpf et al., 2020), and ImageNet top-1 accuracy (Figure 2D; Deng et al., 2009). The only exception was that sighted V1 predictivity measured using regression was not correlated with blind V1 score (Figure 2B; *bottom*). In general, similar trends were observed for recall period activity, but the blind V1 scores were not significantly above baseline.

## 3.3 DNN-BLIND V1 ALIGNMENT IN RESNET50

We then specifically examined ResNet-50 (He et al., 2016), a model known for its shared representations with neural activity in V1 of sighted primates (Schrimpf et al., 2020; Pogoncheff et al., 2023) and which predicted blind V1 activity well. Analysis of alignment scores across the 96 neural recording sites (Fig. 3A) showed significant correlations between DNN outputs and neural activities during the initial $200\,\text{ms}$ of stimulation, except for LFP$_\gamma$ (Fig. 3, left), with ESA showing the strongest correlation (average $r = 0.518$). In the post-stimulation phase ($167 - 367\,\text{ms}$ after onset), only LFP$_\theta$, LFP$_\alpha$, and LFP$_\beta$ activities showed significant positive correlations with DNN outputs. During phosphene recall ($0 - 200\,\text{ms}$ after the audio cue), only LFP$_\beta$ activity significantly correlated with the DNN ($p < 0.01$), suggesting that visual representations in LFP activity, even without electrical stimulation, align with findings that alpha activity is influenced by visual imagery and perception (Bartsch et al., 2015; Xie et al., 2020).

## 3.4 TOWARDS UNDERSTANDING NEURON PROPERTIES BY PROBING ALIGNED DNNS

Optimization-based feature visualization has become a key method for interpreting DNNs (Erhan et al., 2009; Mordvintsev et al., 2015; Olah et al., 2017), offering insights into how these networks process visual information. By applying this technique to our dataset of neural recordings and perceived phosphenes from a blind individual, we aim to uncover the functional characteristics of neurons in blind V1, similar to studies in sighted primate V4 (Bashivan et al., 2019). Our approach involves creating a linear model that maps activations from the first layer of ResNet-50 to blind V1, and computing images that maximally activate specific neural sites according to this mapping.

We demonstrate this concept with maximally exciting images (MEIs) for the neural sites most active during each of the 9 phosphenes, (Figure 3B). Generally, the predicted MEIs share visual features

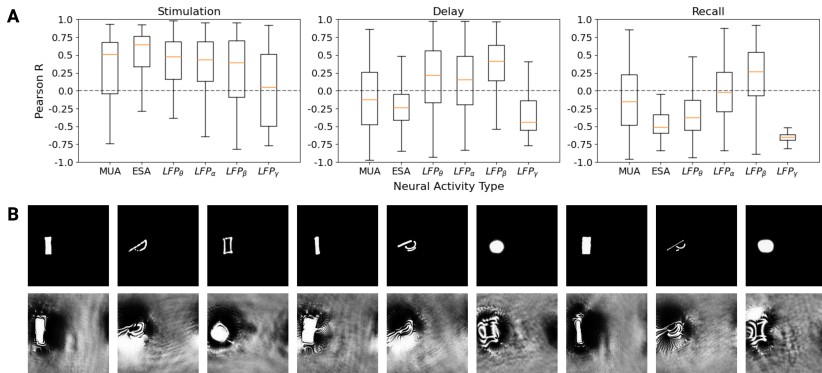

Figure 3: **A**. Blind V1 alignment scores over the 96 neural recording sites during stimulation (left), delay (middle), and recall (right). **B**. Top row: Rendered images of the 9 unique phosphenes perceived by the participant. Bottom row: maximally exciting images (MEIs) predicted for the neuron site that responded with greatest firing rate (relative to it's average activity) for each phosphene.

with the phosphene that elicited a strong response at that site. Given our limited sample, we are hesitant to conclude that these MEIs fully capture the neurons' visual representations, but rather, suggest that this technique may help enhance understandings of visual processing in blind V1.

## 4 DISCUSSION

In this work, we present exploratory analyses on the representational alignment between neural activity measured in V1 of a blind human and latent DNN activations when presented with the same visual perception. While previous research has quantified representational alignment between DNNs and neural activity in V1 of sighted primates, this work is the first of its kind to extend these studies to intracortical neural activity in human, let alone in a blind participant.

Studying blind visual cortex presents unique challenges due to the absence of visual input. To navigate this, we utilized neural responses from electrical stimulation and mental imagery, discovering significant correlations between the activity in blind V1 and both the activity in sighted primate V1 and the performance of DNNs on ImageNet. Specifically, ResNet-50 activations showed strong correlations with neural spiking and various LFP responses during stimulation, with fewer correlations during delay and recall. This opens up future opportunities to delve deeper into these distinct neural activities across different phases. Moreover, we introduce a proof of concept for interpreting blind V1 neurons, suggesting that with a diverse dataset of neural recordings and phosphenes, we could better understand neuron properties and refine stimulation strategies for visual prostheses.

The significant, positive correlation between DNN-blind V1 alignment and DNN-sighted neural alignment suggests the potential use of V1-aligned DNN models for studying visual processing in both sighted and blind individuals. Interestingly, however, we observed that DNN-blind V1 alignment was more correlated with DNN-sighted overall neural predictivity than with DNN-sighted V1 neural predictivity. This observation raises two critical questions: 1) whether electrically evoked visual perception in blind individuals activates different processing mechanisms than natural visual processing, and 2) whether cortical reorganization alters the functional role of V1 in the blind.

A notable limitation of the present study is that the dataset used for analysis contains a limited number of neural responses and visual percepts. Furthermore, our method relies on phosphene drawings for predicting neural activity. Even in this limited setting, well-aligned DNNs could inform more natural stimulation strategies for visual prostheses or deepen our understanding of neural representations in blindness. Moreover, innovative approaches such as topographic networks (Schrimpf et al., 2024) might predict neural activity without these drawings. Despite these challenges, our findings provide a valuable proof of concept, suggesting that existing techniques for assessing representational alignment could be extended and applied to understanding the visual cortex in blind humans.

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

# A  APPENDIX

## A.1  SIGNAL PROCESSING

**Stimulation Artifact Removal**

During the anodic and cathodic pulses of electrical stimulation, noise artifacts distort the underlying neural recordings signals. Signal blanking and filtering are commonly processing approaches applied to prevent such artifacts from corrupting the underlying data (Culaclii et al., 2018). We apply both methods in our analyses of blind V1 representations during stimulation. Hardware-based blanking prevents signal saturation by disconnecting recording electrodes from the amplifier while current is being delivered. As a result, no signal is sampled during these blanking periods. To compensate for this, we performed 3rd order polynomial interpolation within these blanking periods. Finally, signal filtering was applied before computing MUA, ESA, or LFP activity (detailed below).

**Computing Entire/Envelope Spiking Activity (ESA)**

Entire spiking activity is a threshold-free approach to approximating multi-unit spiking activity (Drebitz et al., 2019). In this work, we computed ESA for each channel of recorded neural data by first applying a zero-phase bandpass filter ($750\,\mathrm{Hz}$ low frequency cutoff, $7500\,\mathrm{Hz}$ high frequency cutoff), full-wave rectifying the band-pass filtered signal, applying a zero-phase lowpass filter ($12\,\mathrm{Hz}$ cuttoff frequency), binning the resulting signal into $10\,\mathrm{ms}$, and finally convolving with a Gaussian filter ($25\,\mathrm{ms}$ kernel standard deviation).

**Computing Multi-Unit Spiking Activity (MUA)**

Prior to estimating the multi-unit spiking activity, a zero-phase bandpass filter ($500\,\mathrm{Hz}$ low frequency cutoff, $5000\,\mathrm{Hz}$ high frequency cutoff) was applied to each of the 96 channels of recorded neural data. Subsequently, for each channel, we detected spikes as peaks in the filtered signal that crossed a threshold set equal to 3 standard deviations above the mean peak amplitude of the filtered signal. A peri-stimulus time histogram was then computed from the extracted multi-unit spike times ($10\,\mathrm{ms}$ bin size) and smoothed with a Gaussian filter ($25\,\mathrm{ms}$ kernel standard deviation).

**Computing Local Field Potentials (LFP)**

Theta ($4-8\,\mathrm{Hz}$), alpha ($8-13\,\mathrm{Hz}$), beta ($13-30\,\mathrm{Hz}$), and gamma ($30-100\,\mathrm{Hz}$) local field potentials were studied in this work. Time-series activity in each frequency band was computed by bandpass filtered (zero-phase, $0.5\,\mathrm{Hz}$ low frequency cutoff, $100\,\mathrm{Hz}$ high frequency cutoff) the neural data, computing the time-frequency representation of each channel using a Morlet wavelet transform, and finally averaging the activity in each frequency band within $10\,\mathrm{ms}$ windows.

## A.2  LINEAR FITTING

In order to fit the regression model used to measure blind V1 predictivity, we first projected DNN activations onto the first 90 principal components. Then, cross validated ridge regression was used to learn the mapping between DNN activations and recorded neural activity, using a leave-one-electrode-out cross validation approach. *i.e.*, the data corresponding to all repeat trials of one stimulating electrode on a given day were left out of training, the model was fit to all other data, and predictions were made for the left out activity. This was then repeated across all electrodes for all days, and the correlation (Pearson r) across the left out set was computed as the blind V1 alignment score.

## A.3  FEATURE VISUALIZATION

Visualization of blind V1 neuron preferences was performed using the python package 'lucent' (`github.com/greentfrapp/lucent`). Computing a MEI for a given neuron involved approximating the image that would lead the linearly-mapped DNN to predict a maximum firing rate for the given neuron in question, iteratively computed via gradient descent. Regularization was employed to prevent high frequency artifacts from distoring the resulting MEIs (Mordvintsev et al., 2015; Øygard, 2015; Olah et al., 2017). Specifically, during optimization, the input image was transformed with random jitter, scaling, and rotation at each optimization step. Given that all visual percepts lacked color in this study, we enforced each MEI to be grayscale.

