# OpenReview forum: "Beyond Sight: Probing Alignment Between Image Models and Blind V1"
_ICLR.cc/2024/Workshop/Re-Align — ICLR 2024 Workshop Re-Align ContributedTalk_

### Official Review · Reviewer_R2LX · 2024-02-21
**Cool Idea**

**Rating:** 3
**Fit:** 3
**Confidence:** 3

**Workshop Review:**

This is a really neat setup to study V1 processing in blind subjects. There is the surprising finding that neural activity after electric stimulation can be predicted from DNNs processing the drawn phosphenes. I have a few small comments and questions:

Fig1: Can you connect the image of the phosphene that is drawn and the image that is fed into the DNN? Those are the same, right?

How many total phosphenes, trials and subjects did you use? What is the split into training and test data for the regression models?

2.3 last paragraph, typo"="

3.2. what does 'imagined activity' mean? recall?

It is a little confusing that Fig2A bottom is on recall, but then all of B,C,D is on stimulation, right?

3.3 These are some great questions, but I think there is too much going on. You are juggling: 1) V1 activity in blind 2) V1 activity during WM/recall 3) LFP vs spike coding. All of these are interesting, but, for simplicity and to get tractable scientific insights maybe try to condense the complexity to single hypotheses along one of those dimensions. Imho, question 1) alone is already super exciting.

I really like the last sentence in 3.4, great self-reflection!

I am not sure what to make of the MEIs. They look like the optimization is severely under-constraint?

Here is some prior work on inferring response properties without visual inputs: https://iovs.arvojournals.org/article.aspx?articleid=2773648

**Reason For Not Giving Higher Score:**

n/a

**Reason For Not Giving Lower Score:**

Great idea, well-written, clean execution, good job!

**Reviewer Domain:**

neuroscience

---

### Official Review · Reviewer_Hvvg · 2024-02-23
**Blind human V1 alignment with DNNs**

**Rating:** 3
**Fit:** 3
**Confidence:** 3

**Workshop Review:**

The authors present analyses on shared representations between DNN visual representations (corresponding to drawn phosphenes) and evoked activity in V1 from a blind human. Evoked activity comes from either electrical stimulation or mental imagery. They find that DNNs with better ImageNet accuracy are more predictive of blind V1 activity. Further they provide proof-of-concept for interpreting blind V1 neurons using ResNet-50 alignment. Such results suggest that there is natural visual processing in blind V1 during evoked perception and presents new mechanistic interpretability directions.

The contributions of the paper seem clear and concrete: previous influential work has shown correspondence between human neural activity in visual regions to the latents of pretrained image recognition DNNs, and this work extends such work to intracortical activity and further is the first to do so with a blind participant.

There are currently zero details on the single subject used in this study except for the description that they are blind. What is nature of the subject’s blindness? Was the subject born blind or did such blindness develop during their lifespan? Also are they completely 100% blind or is there any partial visual information that they can experience (e.g., vague brightness levels)? How old was the participant?

For Figure 2B, why do you think regression showed no statistically sig. positive correlation whereas RSA did? Is there an advantage to RSA that can reveal such relationships that regression cannot?

The authors report that LFP-beta activity significantly correlated with DNN and thus visual representations were present even without any electrical stimulation. But it seems like there was significant anti-correlation in the other feature sets, how do the authors explain this and is the LFP-beta significance computed with correction for multiple comparisons?

MEI is never defined in the paper, it took me awhile to guess that it probably stands for maximally evoked images.

**Reason For Not Giving Higher Score:**

N/A

**Reason For Not Giving Lower Score:**

Clear and concrete results, aligns with the workshop theme, that I think has potential for lively discussion.

**Reviewer Domain:**

cognitive science

---

### Official Review · Reviewer_fzmx · 2024-02-24
**Exposing an Interesting and Important New Domain**

**Rating:** 2
**Fit:** 3
**Confidence:** 3

**Workshop Review:**

This paper evaluates the alignment between ANNs and the activities of V1 neurons in a blind subject during/following electrical stimulation.

Strengths:
- The subject is highly novel and potentially very important to understand for medical applications such as retinal prostheses.
- The weaknesses of the approach are both understandable and clearly laid out in the discussion.
- The conclusions, while not necessarily unexpected, seem rigorous, and the inclusion of null results (i.e. the bottom panel of figure 2b) strengthens some of the other claims in my opinion.

Weaknesses:
- There exist strong models for V1 in sighted primates that are well understood and do not require learning (i.e. steerable pyramid responses rank highly for many V1 datasets on the brainscore leaderboard [see here](https://www.brain-score.org/model/vision/1121). This would seem like a much more relevant baseline than the pixel inputs for comparing the deepnet predictivity to.
- The MEI result seems like reading tea-leaves or some sort of memorization thing to me. For example the MEI for multiple phosphenes seem to be swapped. I disagree that this is evidence of a promising approach for gaining understanding of activity in blind V1.
- Reasonable predicitivity is only obtained during the stimulation period (for most forms of neural activity).

Questions:
- It seems quite strange that overall sighted neural predictivity is more correlated with blind V1 predictivity than sighted V1 predictivity. I would appreciate if the authors could speculate on why this might be so.

Overall I think the work explores a potentially very impactful topic, and bringing awareness to this rather wide open problem to the appropriate community seems like a contribution in and of itself. I therefore recommend acceptance.

**Reason For Not Giving Higher Score:**

Some of the experiments are not scientifically very convincing (the MEI results), and I feel the work is still rather preliminary.

**Reason For Not Giving Lower Score:**

The paper explores a highly relevant and potentially quite important topic.

**Reviewer Domain:**

neuroscience

---

### Decision · Program_Chairs · 2024-03-02

Accept (Contributed Talk)